# Quantitative modeling predicts mechanistic links between pre-treatment microbiome composition and metronidazole efficacy in bacterial vaginosis

Christina Y. Lee [1,8], Ryan K. Cheu[2,3,8], Melissa M. Lemke [1], Andrew T. Gustin [2,3], Michael T. France[4], Benjamin Hampel[5], Andrea R. Thurman[6], Gustavo F. Doncel[6], Jacques Ravel [4], Nichole R. Klatt [2,3,7,9✉] & Kelly B. Arnold [1,9✉]

Bacterial vaginosis is a condition associated with adverse reproductive outcomes and characterized by a shift from a *Lactobacillus*-dominant vaginal microbiota to a polymicrobial microbiota, consistently colonized by strains of *Gardnerella vaginalis*. Metronidazole is the first-line treatment; however, treatment failure and recurrence rates remain high. To understand complex interactions between *Gardnerella vaginalis* and *Lactobacillus* involved in efficacy, here we develop an ordinary differential equation model that predicts bacterial growth as a function of metronidazole uptake, sensitivity, and metabolism. The model shows that a critical factor in efficacy is *Lactobacillus* sequestration of metronidazole, and efficacy decreases when the relative abundance of *Lactobacillus* is higher pre-treatment. We validate results in *Gardnerella* and *Lactobacillus* co-cultures, and in two clinical cohorts, finding women with recurrence have significantly higher pre-treatment levels of *Lactobacillus* relative to bacterial vaginosis–associated bacteria. Overall results provide mechanistic insight into how personalized differences in microbial communities influence vaginal antibiotic efficacy.

---

[1] Department of Biomedical Engineering, University of Michigan, Ann Arbor, MI, USA. [2] University of Miami Department of Pediatrics, University of Miami, Miami, FL, USA. [3] Department of Pharmaceutics, University of Washington, Seattle, WA, USA. [4] Institute for Genome Sciences and Department of Microbiology and Immunology, University of Maryland School of Medicine, Baltimore, MD, USA. [5] Division of Infectious Diseases and Hospital Epidemiology, University of Zurich, Zürich, Switzerland. [6] CONRAD, Eastern Virginia Medical School, Norfolk, VA, USA. [7] Department of Surgery, University of Minnesota, Minneapolis, MN, USA. [8] These authors contributed equally: Christina Y. Lee, Ryan K. Cheu. [9] These authors jointly supervised this work: Nichole R. Klatt, Kelly B. Arnold. ✉email: klat0037@umn.edu; kbarnold@umich.edu

Bacterial vaginosis (BV) is a condition that affects 30–60% of women worldwide[1,2], with negative outcomes including increased susceptibility to sexually transmitted infections (STIs) and greater likelihood for adverse reproductive outcomes[3–7]. BV is characterized by a shift from *Lactobacillus* species (spp.)-dominated vaginal microbiota to a diverse array of anaerobic bacteria including *Gardnerella vaginalis* (*Gv*) and *Atopobium vaginae*[8–10]. Treatment of symptomatic BV with metronidazole (MNZ) aims to restore *Lactobacillus*-dominated microbiota; however, recurrence rates remain high, occurring in 57–90% of women who receive treatment[11–14]. Recurrence is associated with several host factors including previous episodes of BV, douching, and sexual activity, but no one factor emerges as a single driver of treatment failure[11,15–18]. In addition, associations between vaginal microbiota composition and BV recurrence have been reported but remain poorly understood, with several studies citing conflicting results[15,17,19].

Recent improvements in 16S rRNA sequencing have enhanced the ability to identify and more accurately quantify the composition of the vaginal microbiota in BV[20,21], finding that the transition is frequently associated with an abundance of *Lactobacillus iners* (*Li*)[22,23]. Despite the association between *Li* and incidence of BV, identifying how *Li* dictates communities of optimal and non-optimal microbiota remains elusive, as the vaginal microbiota can change significantly over time and vary between women[24–26], especially in the presence of MNZ. The recommended treatment regimen for BV consists of oral or vaginal MNZ oriented toward selectively targeting anaerobic bacteria with little effect on *Lactobacillus* spp.[27,28], but high variability in efficacy indicates that further study is required to understand the reestablishment of optimal vaginal microbiota ecosystems.

Recent research in the HIV microbicide field has highlighted the importance of vaginal microbiome composition in drug treatment efficacy. In a landmark study, variability in tenofovir (TFV) microbicide efficacy was accounted for by differences in the vaginal microbiome, specifically the presence of the non-target species *Gv*, which were shown to metabolize TFV[29]. Likewise for MNZ treatment of *Trichomonas vaginalis*, a proposed mechanism of treatment failure was decreased bioavailability of MNZ due to the absorbance of the antibiotic by other microorganisms in the vagina[30–32]. In the context of BV, it is difficult to discern the role of multiple possible interactions that have the potential to influence MNZ efficacy, including MNZ metabolism, resistance, and sequestration across multiple bacterial species that vary considerably among women. We propose that variability in MNZ efficacy may result from underlying differences in MNZ uptake and susceptibility in target and non-target species, and therefore would be highly dependent on individual differences in pre-treatment vaginal microbiota composition.

In this work, we use an ordinary differential equation-based (ODE) model and experimentally measure parameters (MNZ internalization by bacteria, metabolism, and bacterial antibiotic susceptibility) to predict *Li* and *Gv* growth dynamics with MNZ treatment. The model demonstrates that a critical factor in MNZ efficacy may be *Li* sequestration of MNZ, and predicts that MNZ efficacy decreases in individuals with higher pre-treatment amounts of the non-target species, *Li*, relative to the target species, *Gv*. We validate this finding with in vitro co-cultures, and extend our analysis to more representative models which illustrate that this behavior is also expected in microbial environments with additional species, interspecies interactions, and strain variability. Finally, by analyzing cervicovaginal samples from BV-infected women treated with MNZ in two distinct cohorts we demonstrate that our initial findings have clinical relevance in characterizing BV treatment outcomes[33,34]. Overall, our findings highlight the importance of leveraging quantitative models that evaluate interactions of target bacteria and non-target *Lactobacillus* spp. with MNZ in improving insight into personalized differences in BV recurrence and treatment failure.

## Results

**Model predicts *Lactobacillus* MNZ sequestration influences efficacy.** To determine how MNZ treatment efficacy can be altered by bacterial-mediated interactions in vitro, we created an ODE model to predict growth of *Gv* and *Li* upon co-culture and treatment with MNZ (Fig. 1a). Parameters for each bacterial species were obtained by least-squares fitting of in vitro kinetic data and dose–response curves for MNZ exposure with each species in monoculture (Supplementary Figs. 1, 2, Supplementary Table 1), before the ODE model was used to predict co-culture conditions with *Gv* and *Li* both interacting with extracellular MNZ. The model assumes that *Gv* and *Li* internalize or sequester MNZ at rates $k_{\text{int-Gv}}$ and $k_{\text{int-Li}}$, respectively, and *Gv* can convert MNZ to the stable metabolite, acetamide, and unknown metabolites at rate $k_{\text{met}}$[35]. The model additionally assumes logistic growth at rates $k_{\text{grow-Gv}}$ and $k_{\text{grow-Li}}$, with carrying capacities of $K_{\text{Gv}}$ and $K_{\text{Li}}$ and growth inhibition by MNZ toxicity at rates $k_{\text{kill-Gv}}$ and $k_{\text{kill-Li}}$ in a dose-dependent manner based on 50% effective concentrations of MNZ on *Gv* and *Li* (EC50$_{\text{Gv}}$, EC50$_{\text{Li}}$)[36,37]. Since MNZ is a pro-drug that is activated when internalized by anaerobic bacteria, the cytotoxicity of MNZ in the model is dependent on the intracellular concentration of MNZ rather than extracellular MNZ concentration; however, we used the external MNZ concentration as the basis for EC50 of internalized MNZ, as experimentally determining the intracellular level of MNZ per cell was challenging and the main goal was to capture the relative sensitivity between *Gv* and *Li*[27,28,30,38].

To identify model parameters that were most critical for decreasing *Gv* growth, we performed a 1-dimensional (1D) sensitivity analysis by altering each parameter three orders of magnitude above and below baseline and evaluated *Gv* growth (Fig. 1b, c). Growth was scaled relative to the predicted growth in an unperturbed (no MNZ) co-culture based on the time point and initial population sizes evaluated and is referred to as percent maximum growth. The sensitivity analysis identified *Gv* growth as highly dependent on the MNZ internalization/sequestration rate into *Li* ($k_{\text{int-Li}}$). A 50-fold increase in this rate increased the growth of *Gv* from 7.42 to 69.5% its maximal growth upon 48 h treatment with MNZ (Fig. 1c). Likewise, changing the MNZ internalization rate into *Gv* ($k_{\text{int-Gv}}$) has similar effects on *Li* where increasing this rate 50-fold resulted in 89.7% *Li*'s maximal growth (Supplementary Fig. 3a). Overall, these results illustrate how MNZ efficacy in inhibiting *Gv* growth is influenced by the competition between each bacterium to internalize the drug.

From this result, we hypothesized that the relative quantity of cells internalizing MNZ (ratio of *Gv* and *Li*) could significantly influence growth of both strains. We tested this hypothesis in our computational framework by predicting *Gv* survival after varying the starting ratio of *Gv* to *Li* (Gv:Li ratio) from 1000× fold to 0.001×. Results indicated that altering the initial Gv:Li ratio influences the growth of both *Gv* and *Li*. Counterintuitively, *Gv* survival was high when *Li* initially outnumbers *Gv* 1000× and *Li* growth is optimal when *Gv* initially exceeds *Li* 1000× (Fig. 1d). Stated differently, the model suggested that more *Gv* present at MNZ treatment initiation resulted in a better treatment outcome. The importance of MNZ internalization rate into *Li* became more apparent as *Li* became the predominating species, leading to increased growth of *Gv* (Fig. 1e). This result additionally supports

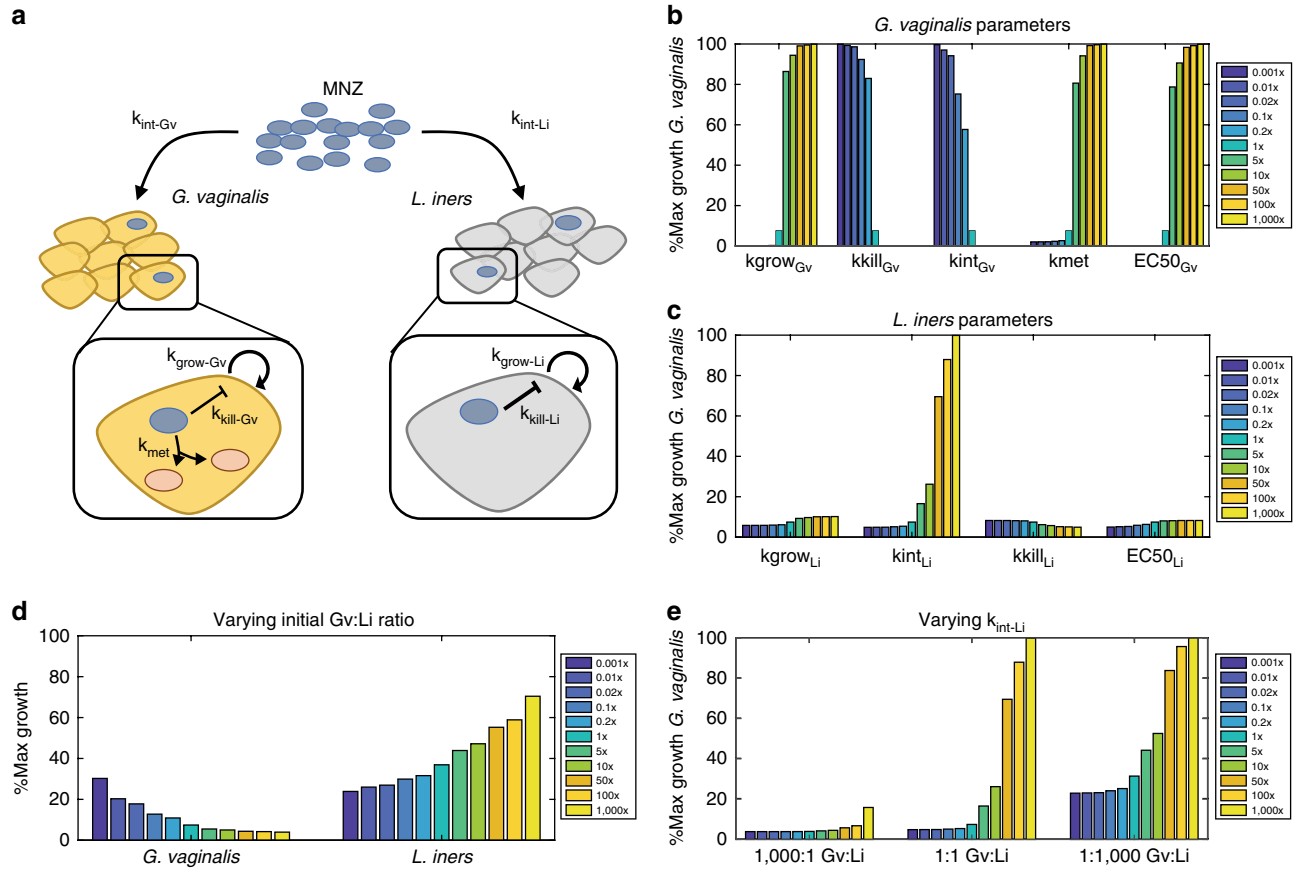

**Fig. 1 Model schematic for bacterial growth dynamics in BV with MNZ treatment. a** MNZ is internalized by both *G. vaginalis* (*Gv*) and *L. iners* (*Li*) at rates $k_{int-GV}$ and $k_{int-LI}$, cells are proliferating at $k_{grow-GV}$ and $k_{grow-LI}$ and MNZ inhibits growth by $k_{kill-GV}$ and $k_{kill-LI}$. For *Gv*, a potential mechanism of MNZ resistance is the bacterial-mediated interactions to the drug leading to the formation of metabolites ($k_{met}$). **b** Sensitivity of *Gv* growth with 500 μg/ml MNZ when parameters directly related to *Gv* growth are varied 0.001× to 1000× baseline values. Percent maximal growth refers to the final cell count compared to the carrying capacity of the culture, or the maximum cell density the unperturbed culture can reach at 48 h based on initial cell density. **c** Sensitivity of *Gv* growth with 500 μg/ml MNZ when parameters related to *Li* survival are varied 0.001× to 1000× baseline values. **d** Percent maximal growth of *Gv* (left) and *Li* (right) when the initial ratio of *Gv* to *Li* is varied with 500 μg/ml MNZ treatment. **e** Percent maximal growth of *Gv* when MNZ internalization rate of *Li* is varied at three different population compositions with 500 μg/ml MNZ treatment.

the concept that *Li* competes with *Gv* to internalize or sequester extracellular MNZ, as when one bacterial strain is in excess, it likely depletes available extracellular MNZ and decreases the amount of drug internalized by the non-dominating bacterial strain.

We used our model to explore this ratio-dependent behavior over a range of relevant MNZ concentrations extending from 100 to 1600 μg/ml, as estimates for vaginal accumulation range from 20 to >1000 μg/ml (Fig. 2a, b)[39,40]. Doses below 100 μg/ml had no effect on *Gv* or *Li* growth and doses above 1600 μg/ml exhibited near-complete cell killing for both bacterial strains (Fig. 2a, b); these data are in agreement with experimentally determined effective concentrations of MNZ on *Gv* and *Li* cultured individually (Supplementary Fig. 2c, d). However, for doses between 100 and 1600 μg/ml there were significant differences depending on the initial Gv:Li ratio, where MNZ was most efficacious in eliminating *Gv* when more *Gv* than *Li* was present initially.

### Model validation in *Gv* and *Li* co-cultures

We validated these counterintuitive model predictions experimentally in vitro by varying the initial Gv:Li ratios in the presence of 500 μg/ml MNZ

and tracking growth for 48 h (Fig. 2c, d). Experimental measurements confirmed model predictions that MNZ efficacy for inhibiting *Gv* growth decreased when *Li* was initially dominant ($p = 4.67 \times 10^{-8}$), and were not significantly different than model predictions (0.001× Gv:Li, $p = 0.430$; 1000× Gv:Li ratio, $p = 0.689$, Fig. 3c), with *Gv* exhibiting a predicted 30.3% and experimental 41.4% ± 13.3% maximal growth after treatment when *Li* was initially dominant compared to a predicted 2.1% and experimental 9.4% ± 13.8% maximal growth when *Gv* initially was dominant. *Li* growth in the presence of 500 μg/ml MNZ was also dependent on the initial Gv:Li ratio, where MNZ inhibited *Li* growth the most when *Li* was initially dominant, 7.2% ± 3.9% maximal growth compared to when *Gv* was initially dominant, 70.5% ± 33.8% ($p = 3.29 \times 10^{-9}$, Fig. 2d). Notably, the model over-predicted the growth of the *Li* population when *Li* was initially dominant (0.001× Gv:Li), where the model prediction of 23.9% maximal growth was over threefold higher than experimentally observed, 7.19% ± 3.91% growth (0.001× Gv:Li experiment vs simulation, $p = 1.43 \times 10^{-5}$), suggesting efficacy dependence on a high pre-treatment Gv:Li ratio may be even greater than that predicted by the model. Experimental and model predictions of *Li* growth were not significantly different when *Gv* was initially dominant (1000× Gv:Li, $p = 0.726$).

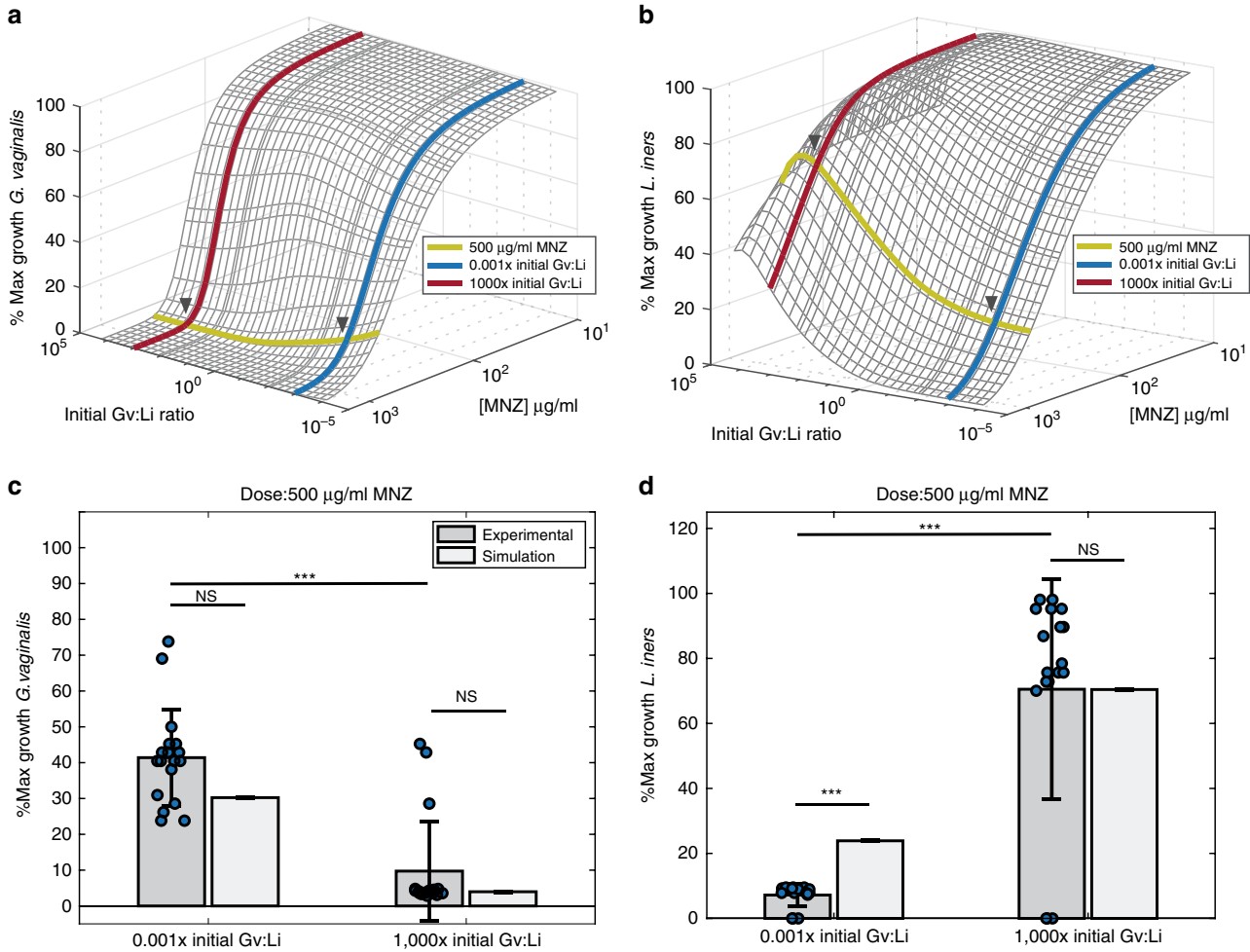

**Fig. 2 A higher initial Gv:Li ratio improves MNZ treatment efficacy. a** Surface plot to illustrate predicted percent maximal growth of *Gv* (*z*-axis) when concentration of MNZ (*x*-axis) and the ratio of Gv:Li (*y*-axis) are varied in simultaneously. Arrows indicate the concentration of MNZ and ratios of Gv:Li used for model validation. **b** Percent maximal growth of *Li* after simultaneous variation of MNZ dose and Gv:Li ratio. **c, d** Comparison of model simulations to experimental data for 500 µg/ml MNZ at 1000× and 0.001× Gv:Li. *Gv* percent maximal growth 0.001× initial Gv:Li ratio and 1000× initial Gv:Li ratio experimental vs simulation, and experimental vs experimental *p*-values were $p = 0.430$, $t = 0.809$, df = 17; $p = 0.680$, $t = 0.420$, df = 17; $p = 4.67 \times 10^{-8}$, $t = 6.99$, df = 34, respectively. *Li* percent maximal growth 0.001× initial Gv:Li ratio and 1000× initial Gv:Li ratio experimental vs simulation, and experimental vs experimental *p*-values were $p = 1.43 \times 10^{-5}$, $t = 6.00$, df = 17; $p = 0.726$, $t = 0.357$, df = 17; $p = 3.29 \times 10^{-9}$, $t = 7.91$, df = 34, respectively. Data are presented as mean ± SD, $n = 18$ independent, biological replicates for each initial ratio, asterisks indicate significance as: *$p < 0.05$, **$p < 0.01$, ***$p < 0.001$ without adjustment for multiple comparisons, unpaired two-sided *t*-test. Source data are provided as a Source data file.

Likewise, model predictions of MNZ and MNZ metabolite concentrations were not significantly different from experimental results in cultures starting with a 0.001× Gv:Li ratio (extracellular MNZ: $p = 0.255$, intracellular MNZ: $p = 0.336$, acetamide: $p = 0.877$), but predictions for extracellular MNZ, intracellular MNZ, and acetamide concentrations in cultures with a 1000× Gv:Li ratio did vary significantly from experimental data (Supplementary Fig. 4). The deviation of model predictions when *Gv* is initially dominant suggests that experimental investigation of detailed mechanisms of *Gv* interactions with MNZ is warranted (for example, the potential ability of *Gv* to externally degrade MNZ). Despite some deviation of peripheral model predictions from experimental measurements, the Gv:Li ratio-dependent trends were reproduced by the model. The dependency on initial culture ratios of *Gv* to *Li* on growth suggests that non-target bacteria that sequester MNZ could significantly alter drug efficacy.

We observed some variation in the sensitivity (EC50) of *Li* to MNZ. Variability in minimum inhibitory concentrations (MIC) estimations have been reported, as changes in culture conditions

including incubation length and the inoculum effect can influence the apparent sensitivity of bacteria to antibiotic[37,41]. In addition, the sensitivity of *Lactobacillus* spp. and *Gv* to MNZ and their MICs are reported to range from 500 to 4000 µg/ml and 0.75 to >256 µg/ml, respectively[42–44]. To ascertain whether our results would be influenced by variation in *Li* sensitivity to MNZ, we repeated the simulations over a range of EC50 values. To represent reported resistance of *Lactobacillus* spp. in vitro, we increased the EC50 value of *Li* to be 10-fold higher than *Gv* (EC50$_{Li}$ = 4200 µg/ml). MNZ efficacy in inhibiting *Gv* growth was similarly decreased at low Gv:Li ratios (36.5% max growth at 0.001× Gv:Li) compared to high Gv:Li ratios (3.96% max growth at 1000× Gv:Li, Supplementary Fig. 5a). *Li* had little to no susceptibility over the range of MNZ concentrations tested (Supplementary Fig. 5b). In addition, these EC50 values replicated trends in experimental data for growth kinetics (Supplementary Fig. 2d, h versus Supplementary Fig. 5c, d). These results support that the initial Gv:Li ratio-dependent trends in MNZ efficacy for inhibiting *Gv* growth are independent of *Li*'s sensitivity to MNZ.

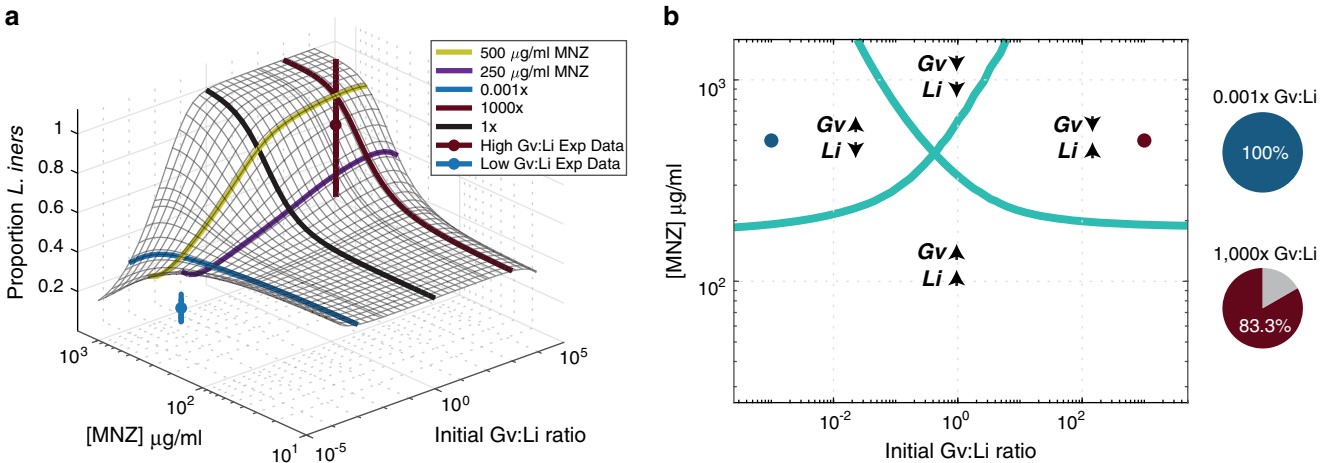

**Fig. 3 Initial Gv:Li ratios dictate final microbial populations. a** Surface plot illustrates model predictions for proportion of *Li* relative to *Gv* 48 h at different starting Gv:Li ratios (x-axis) and at different doses of MNZ (y-axis). Experimental validation was performed in in vitro co-cultures of *Li* and *Gv* (n = 18 independent, biological replicates for each ratio) and is plotted on the surface, with mean ± SD represented by nodes and vertical lines. **b** Phase diagram of microbial growth dynamics 48 h after exposure to various MNZ doses, dots indicate experimental conditions evaluated. There are four possibilities: Both *Gv* and *Li* populations are increased after treatment, both *Gv* and *Li* populations are decreased, only the *Gv* population is increased and only the *Li* population is increased. Pie charts indicate the fraction of experimental samples that agree with the predicted trends (right).

**Optimal MNZ doses are dependent on pre-treatment microbiome**. We next used the model to determine specific combinations of MNZ concentrations and initial Gv:Li ratios that resulted in optimal final *Li* proportion after 48 MNZ exposure. The initial Gv:Li ratio was highly associated with the final Gv:Li ratio for doses of MNZ >250 μg/ml (Fig. 3a). Interestingly, cultures that were initially *Li* dominant (0.001× Gv:Li), were nearly insensitive to any dose of MNZ, resulting consistently with >50% *Gv* (Fig. 3a). This result carries the surprising implication that women with *Li*-dominant vaginal microbiomes at treatment initiation are likely to undergo recurrence, regardless of MNZ dose. Of note, cultures that were originally *Gv* dominant (Gv:Li >1) were the most likely to be *Li* dominated after 48 h exposure to MNZ. Experimental data supported these trends, as the simulation predictions were not significantly different for the final proportion of *Li* at 500 μg/ml for 1000× (p = 0.680, t = 0.420, df = 17). The model did overestimate the final proportion of *Li* at the 0.001× Gv:Li ratio (predicting a 44.1% proportion of *Li* compared to 14.2% ± 7.16% obtained experimentally); however, this result suggests an even more significant reduction in *Li* proportion when *Gv* is initially dominant (p = 0.008, t = 4.06, df = 17).

A phase diagram of MNZ therapy outcomes at 48 h was created to characterize both *Li* and *Gv* endpoint growth dynamics, which depict either an increase/expansion or decrease in population size relative to the initial population. The optimal growth dynamics would depict the expansion of only the *Li* population and the least optimal growth dynamics would be the expansion of only *Gv*. A decrease in both populations is additionally not optimal, as lower levels of beneficial microbiota are often associated with opportunistic infections or overgrowth of non-optimal species[45,46]. We observed that higher initial Gv:Li ratios in conjunction with MNZ concentrations over 250 μg/mL were more likely to result in optimal final growth dynamics where the *Li* bacterial population was the only population expanding (Fig. 3b). Likewise, it was possible for only the *Gv* population to grow and the *Li* population to decrease when the initial Gv:Li ratio was <1×. Interestingly, the diagram predicts that it is possible that both *Gv* and *Li* populations would decrease for intermediate ratios of Gv:Li, which expand to include a wider

range of ratios as the dose of MNZ is increased. Overall, in vitro co-culture experimental data supported the model predictions for endpoint growth dynamics, with 15 of 18 samples agreeing with the dynamics predicted by the phase diagram for the 1000× Gv:Li, 500 μg/ml group and for all 18 samples agreeing with the predictions for the 0.001× Gv:Li ratio, 500 μg/ml group (Fig. 3b, right). This result reinforces the importance of pre-treatment Gv: Li ratio on post-treatment bacterial community composition.

**Initial composition influences efficacy in more complex models**. While our model results emphasize the importance of pre-treatment Gv:Li ratios in MNZ efficacy in co-cultures, BV in women is more complex, and involves interspecies interactions and strain variability across many different bacterial species. To evaluate the above results in more complex settings that include multiple species, interspecies interactions, and strain variability, we created three additional model structures (Fig. 4a–d). In Model B and Model D, we account for potential interspecies interactions, such as amensalism between *Lactobacillus* spp. and BV-associated bacteria and commensal or mutualistic behavior within BV-associated bacteria subpopulations and *Lactobacillus* spp. (Fig. 4a, d)[47–51]. In Models C and D we add additional representative species; a second BV-associated species and second *Lactobacillus* spp. (Fig. 4c, d). To address potential variability in associated parameters, we randomly selected parameter values from physiologically relevant ranges determined from previously published studies (Supplementary Tables 2 and 3). Notably, across all four model structures we found that higher initial relative amounts of BV-associated bacteria to *Lactobacillus* spp. had higher relative post-antibiotic levels of *Lactobacillus* spp. (BV: LB ratio, Fig. 4e, f, Supplementary Fig. 6, p < 1E–6, p < 1E–6, p < 1E–6, p < 1E–6). This result held for a range of ratios (0.6× BV:LB and 100× BV:LB) chosen to reflect the observed relative abundance of *Lactobacillus* spp. in BV positive women (60–1.0%)[52]. Moreover, for each of these model structures, the global sensitivity analyses consistently selected the MNZ internalization/sequestration parameter ($k_{int}$) and the initial relative abundance of BV-associated bacteria to *Lactobacillus* spp. (BV:LB ratio) as significantly sensitive parameters in post-antibiotic treatment *Lactobacillus* spp. relative abundance. Variability in *Gv* sensitivity

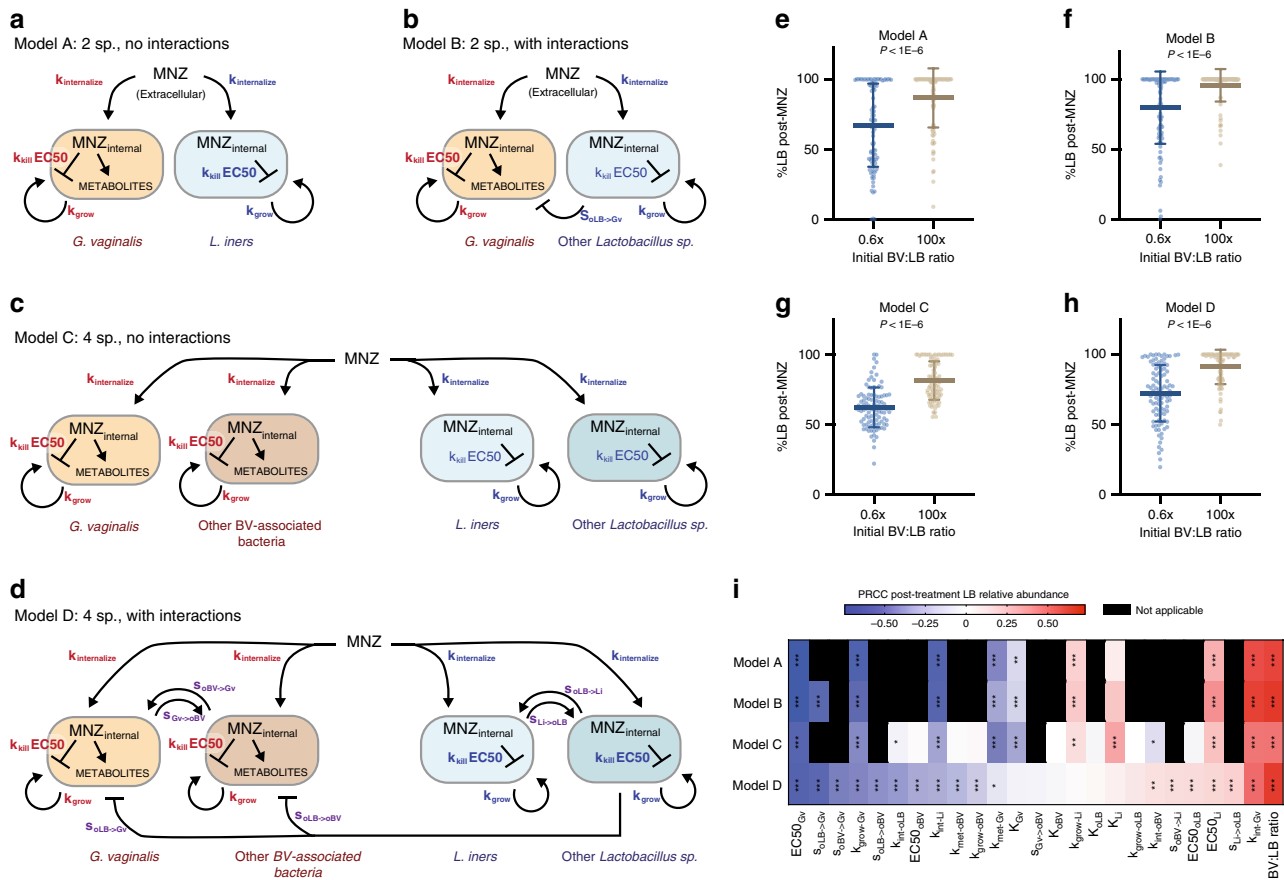

**Fig. 4 High pre-treatment BV:LB ratio is predicted to reduce MNZ efficacy in more complex microbial environments regardless of strain variability.** **a** Original model structure validated in Fig. 2. **b** Two-species model with negative interaction between other *Lactobacillus* spp. (oLB) and *Gv*. **c** Four species model of *Gv* and *Li* with additional representative bacteria for BV-associated bacteria and *Lactobacillus* spp. **d** Four species model with interspecies interactions. Within BV-associated bacteria and *Lactobacillus* spp. interactions were simulated from mutualistic (both benefit) to commensal (one benefits, the other is neutral). Inhibitory (amensal) interactions are included between D-lactic acid producing bacteria, other *Lactobacillus* spp., with both BV-associated bacteria. **e–h** Post-MNZ treatment (48 h, 500 μg/ml) *Lactobacillus* spp. relative abundances at 0.6× and 100× BV-associated bacteria to *Lactobacillus* spp (BV:LB) ratios for each model type (*n* = 100 independent simulations for each ratio, data are presented as mean ± SD). Each point represents a parameter set randomly sampled from physiological ranges in Supplementary Tables 2 and 3. Statistical analysis was completed using unpaired, two-sided *t*-tests: Model A (*p* = 7.20 × 10⁻⁷, *t* ratio = 5.32, df = 198), Model B (*p* = 1.67 × 10⁻⁷, *t* ratio = 5.649, df = 198), Model C (*p* = 8.05 × 10⁻¹⁸, *t* ratio = 9.725, df = 198), and Model D (*p* = 1.70 × 10⁻¹³, *t* ratio = 7.954, df = 198), which were corrected for multiple comparisons using the Benjamini and Hochberg method. **i** Significantly sensitive parameters were assessed by partial rank correlation for each model structure (**a–d**) in a global sensitivity and uncertainty analysis, multiple comparisons where adjusted for using Bonferroni correction (asterisks indicate significance as **p* < 0.05, ***p* < 0.01, ****p* < 0.001). Source data are provided as a Source data file.

to MNZ (EC50) and growth rate were also selected as critical parameters in dictating response to MNZ treatment, which are of interest as there is significant variability across *Gv* subclasses in terms of resistance to MNZ, and metabolism[53]. Furthermore, when models were modified such that *Lactobacillus* spp. could not internalize/sequester MNZ, the ratio-dependent effect was abrogated, and was independent of sensitivity of *Lactobacillus* spp. to MNZ (Supplementary Fig. 7a, b). Altogether, this provides additional quantitative evidence that *Lactobacillus* spp. sequestration of MNZ may contribute to BV recurrence in more complex microbial environments.

**Pre-treatment composition is associated with clinical outcome.** We next evaluated whether the influence of initial BV:LB ratio on MNZ efficacy is observed clinically. We compared the pre-treatment ratio of BV-associated bacteria to *Lactobacillus* spp.

(BV:LB ratio) in vaginal samples collected from women who underwent MNZ treatment for BV and were cured or experienced recurrence, in two clinical studies; the UMB-HMP study[33] (*n* = 11) and CONRAD BV study[34] (*n* = 33). We chose to evaluate each study separately to minimize the effects of differences in sample collection and in methods of microbial species measurements. In the UMB-HMP cohort, 11 women were observed over the course of 10 weeks and provided cervicovaginal lavage (CVLs) samples each day for quantification of relative microbial abundances by sequencing of the V3 and V4 regions of 16S rRNA. Patients underwent treatment for BV that consisted of 1 week of 500-mg oral MNZ, taken twice daily. Of the 11 women, 8 met inclusion criteria and were classified as recurrent or cured dependent on Nugent scoring, where recurrent patients were described as women who responded to treatment but exhibited a second episode of BV during the 10-week period (Supplementary Table 4). Results resonated with model predictions where

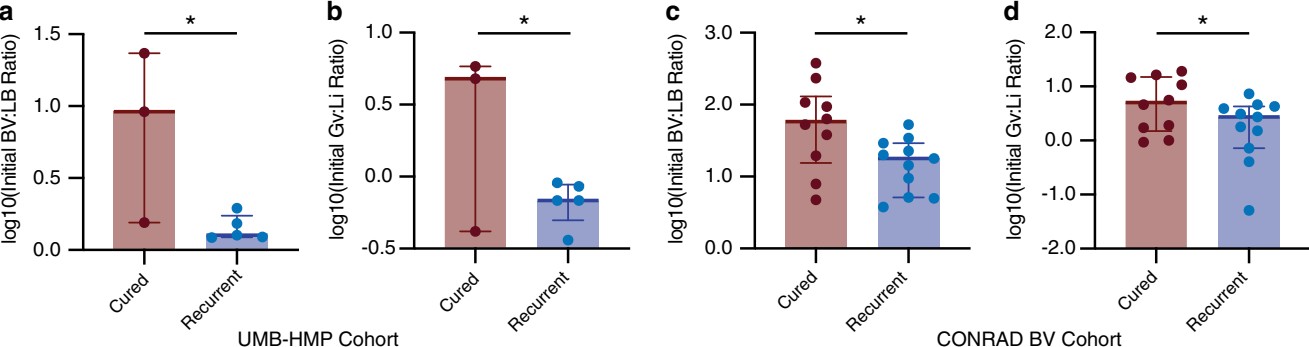

**Fig. 5 Increased initial BV:LB ratios associated with successful treatment of BV. a**, **b** Clinical results for the UMB-HMP cohort ($n = 3$ individuals for the cured group, $n = 5$ individuals for the recurrent group) describing the **a** log base 10 transform of initial BV-associated bacteria relative abundance to *Lactobacillus* spp. relative abundance, $p = 0.0366$, $t = 2.678$, df = 6. **b** Initial Gv:Li ratio, $p = 0.0497$, $t = 2.451$, df = 6. **c**, **d** Clinical results for the CONRAD BV cohort ($n = 10$ individuals for the cured group, $n = 11$ individuals for the recurrent group) describing **c** log base 10 transform of initial BV-associated bacteria relative abundance to *Lactobacillus* spp. relative abundance, $p = 0.0242$, $t = 2.449$, df = 19. **d** Initial Gv:Li ratio, $p = 0.0338$, $t = 2.287$, df = 19. Data are presented as median, 25th and 75th quartiles, statistical analysis was completed with unpaired, two-sided *t*-tests that were not adjusted for multiple comparisons. Source data are provided as a Source data file.

individuals who experienced recurrence had higher amounts of *Lactobacillus* spp. relative to BV-associated bacteria (lower BV:LB ratios, $p = 0.0366$) at treatment initiation and tended to have higher abundances of *Lactobacillus* spp., particularly *Li*, but abundance of individual species were not statistically significant after adjustment for multiple comparisons ($p = 0.201$, Fig. 5a, Supplementary Fig. 8a). In addition, *Gv* relative abundance was not significantly different between groups ($p = 0.984$, Supplementary Fig. 8b). Furthermore, when we analyzed the specific species in the original two-species model, we also observed similar results where cured women had significantly higher ratios of *Gv* to *Li* ($p = 0.0497$, Fig. 5b). It is important to note that since the Gv:Li ratio comparison was a selective analysis, we did not correct for multiple comparisons based on individual species in the original data set (over 190 species measured). These results support both the in vitro experimental data and model results that predicted a lower efficacy of MNZ treatment when a lower ratio of *Gv* to *Li* was present pre-treatment.

We also evaluated model findings in a second clinical cohort, the CONRAD BV study, which consisted of 33 women whose vaginal microbiome was sampled at enrollment in the study, 1 week after MNZ treatment and 1 month after MNZ treatment. Relative abundances were determined by sequencing of the V4 region of the 16S rRNA. Women were excluded from this subset analysis if they failed to finish the antibiotic regimen, contracted a secondary vaginal infection, did not respond or had delayed response of treatment. Of the 33 women, 21 met inclusion criteria and were evaluated by molecular-BV criteria (dominance of *Lactobacillus* spp.) at 1 week and 1 month, with women exhibiting a vaginal microbiota composition of <50% *Lactobacillus* spp. classified as BV positive. The group analyzed consisted of women who were cured ($n = 10$) and or were determined to have recurrent BV ($n = 11$; *Lactobacillus* was dominant at 1 week, but molecular-BV returned after 1 month, Supplementary Table 5). Like the previous study, we found that women who experienced recurrence had higher levels of *Lactobacillus* spp. relative to BV-associated bacteria (lower BV:LB ratio, $p = 0.0242$, Fig. 5c). Comparison of CLR-transformed relative abundance did not result in statistically significant differences for *Li* or *Gv*, but tended to support the trend of recurrent women having higher *Li* and lower *Gv* (Supplementary Fig. 8c, d, $p = 0.521$, $p = 0.694$). Similarly, analysis of the Gv:Li ratio supported higher pre-treatment *Gv* relative to *Li* was associated with better treatment

outcomes (Fig. 5d; $p = 0.0338$). Though preliminary and limited by low sample numbers, these results support the model predictions and suggest that successful BV treatment could be driven by competition for MNZ, where non-target bacterial populations, *Lactobacillus* spp., like *Li* sequester MNZ away from target bacterial populations like *Gv*, *A. vaginae*, *Sneathia*, etc., ultimately decreasing MNZ efficacy.

## Discussion

Here we show a personalized tolerance mechanism that may contribute to BV recurrence and treatment failure. Our model illustrates how non-target bacteria, such as *Li* or other *Lactobacillus* spp., may sequester antibiotic and lower the amount of MNZ available to target bacteria like *Gv*. This model result implies that MNZ efficacy may be dependent on highly variable pre-treatment relative abundances of *Lactobacillus* spp. such as *Li* to BV-associated bacteria populations (BV:LB ratios) and raises the question of whether patients with higher levels of *Lactobacillus* spp. are more susceptible to recurrent BV than those with higher degrees of dysbiosis. Importantly, results from the model, in vitro experiments, and clinical data all point to a higher pre-treatment BV-associated bacteria population relative to *Lactobacillus* spp. as a driver of MNZ efficacy in inhibiting *Gv* growth and facilitating post-treatment Lactobacillus dominance. This study complements ongoing work in the search for drivers of BV treatment efficacy, in which experimental studies are often limited to delineating the role of individual bacteria, and it is challenging to assess the importance of numerous clinical and microbial variables that are associated with treatment outcome[15].

The potential for non-antibiotic-target bacterial populations to act as a sink for MNZ and alter efficacy is similar to a concept that has been previously explored in bacterial ecology, termed the inoculum effect (IE), which describes an increase in antibiotic MICs due to increased bacterial load and decreased per cell antibiotic concentration[54]. While the IE and the ability of bacterial species to influence MNZ bioavailability has been previously reported, to our knowledge its role in BV recurrence has not yet been considered. Furthermore, the ODE model used here was essential for determining the critical importance of MNZ sequestration by *Lactobacillus* spp. across multiple interactions that have the potential to influence efficacy and recurrence, including metabolism, proliferation, and susceptibility to MNZ of both target and non-target species. The model was also necessary

for translating the importance of this parameter to microbial communities with varying compositions and with different MNZ dosing regimens. Though the proposed MNZ sequestration mechanisms were not experimentally validated in this study, the model predictions for associated relationships between pre-treatment microbial composition and BV recurrence were reca-pitulated in both co-cultures and in cervicovaginal samples, providing an additional mechanism for recurrence that has not previously been considered.

Recent studies evaluating pre-treatment vaginal microbiota composition on MNZ efficacy have reported inconsistent results, likely due to differences in patient exclusion criteria, time point of treatment outcome assessment, drug regimen, and methods to collect and quantify the vaginal microbiota. One study that employed a similar drug regimen (oral MNZ) and sample col-lection methods to the clinical cohorts evaluated here supported our results, finding higher pre-treatment loads of antibiotic-target species, Gv and A. vaginae, associated with BV treatment effi-cacy[19]. Other studies that used different sample collection methods and antibiotic regimens did not explicitly evaluate the pre-treatment ratio of BV-associated bacteria to Lactobacillus spp.; generally suggested there was an association between total Lactobacillus relative abundance and successful treatment[15,55,56]. Notably, some of these studies focused on analyzing treatment outcome immediately after antibiotic therapy was completed, and in some cases treatment failure was due to no response to ther-apy. We propose that recurrence and failure to respond to ther-apy likely arise from different factors, where recurrence is due to a collective bacterial population's resilience to antibiotic therapy and failure to respond is due to inherent resistance of BV-associated bacteria. Studies that have associated higher Gv loads with treatment failure correspond with the latter and could be due to the formation of biofilms or other resistance mechan-isms[56]. As our model predicts immediate post-therapy Lactoba-cillus spp. relative abundance, no response to treatment would be equivalent to predicting no change or low Lactobacillus spp. relative abundance at 48 h. An additional limitation of our model is that it does not appear to be applicable to cases of MNZ treatment failure in women who initially had very low levels of Lactobacillus spp. (<1%), which our model would predict should promote MNZ efficacy[34]. However, we propose that treatment failure in this case may be a result of the Allee effect[57,58], which can be caused by a variety of mechanisms that lead to decreased fitness at low population densities, suggesting these women have Lactobacillus abundances that are too low to recolonize the vagina and may be associated with more precisely modeling interspecies interactions. Moreover, since Li is the only Lactobacillus spp. observed to date to significantly sequester MNZ, it will be important to characterize how other vaginal bacterial species interact with MNZ to further explore the role of non-target bacterial species on MNZ efficacy. Altogether, conflicting results in clinical studies of pre-treatment vaginal microbiota composi-tion support the need for the development of quantitative plat-forms to evaluate the interplay between multiple microbial species, clinical variables, and dosing regimens that contribute to personalized differences in treatment failure.

Models presented here are only simple reconstructions of the minimal possible interactions between bacterial species and an antibiotic that have been established as key species by the existing literature[10,15,17], with a time-scale that was limited by in vitro co-culture conditions. While the model provided useful insight into how non-target bacterial species may influence BV recurrence after MNZ treatment, predicting regrowth of Lactobacillus spp., and the full quantitative mechanisms underlying responses to treatment are likely more nuanced. More complex model fra-meworks did suggest key results would hold true in microbial communities with additional microbial species, interspecies interactions, and strain variability, though we were not able to validate this experimentally. Interspecies interactions in our models were incorporated with generalized Lotka–Volterra equations which simplifies relationships to a single term, but represent a good starting point for recapitulating ecosystem-level complexities[59–63]. Specific metabolic interactions that dictate survival and elimination of bacterial species in the vagina could be included with greater mechanistic detail in the future. In instances where parameters are unknown or difficult to measure experimentally, this work demonstrates the value of a global computational sensitivity analysis for understanding the relative importance of strain-level differences in antibiotic uptake, metabolism, or sensitivity. Predictive simulations can be run across multiple possible parameter ranges to determine the effects of variation prior to costly experimental measurements. This tool will be valuable in isolating the role of individual parameters in making a bacterial population or community more tolerant to antibiotic therapy.

In this study, we demonstrated that ODE models can provide insights into antibiotic–microbe interactions pertinent to under-standing BV treatment efficacy. Our work highlights that it is possible for BV treatment to fail, even if target bacteria are not resistant to MNZ as vaginal bacterial populations as a whole can be resilient to antibiotic, resulting in recurrent BV. While our clinical analysis is limited in sample size and therefore should be considered preliminary, future extensions of this work could be used to inform clinical decision-making regarding personalized therapy options. More generally, we envision that the use of quantitative models such as this will provide a framework for integrating knowledge of interactions between multiple bacterial species and drug treatments in mucosal tissues to give insight into the diverse responses observed in infectious disease and other syndromes of the female reproductive tract.

## Methods

**Bacterial strains and culture conditions.** Lactobacillus iners ATCC 55195 and Gardnerella vaginalis ATCC 14018 (group C) were obtained from the American Type Culture Collection (ATCC) and maintained on Human Bilayer Tween Agar (BD) plates and New York City III (NYCIII) medium according to the manu-facturer's instructions. Agar plates and liquid cultures were incubated at 37 °C with anaerobic gas mixture, 80% $N_2$, 10% $CO_2$, and 10% $H_2$. Frozen stocks of strains were stored at −80 °C in 40% (v/v) glycerol.

**Metronidazole quantification by tandem mass spectrometry.** MNZ con-centrations were determined by validated LC-MS/MS assays. Sample aliquots were centrifuged at $3000 \times g$ and divided between supernatant and cell pellet. Extra-cellular MNZ was extracted from supernatant via protein precipitation using acetonitrile. For intracellular concentration measurements, cell pellets were lysed using sonication and re-suspended in 100 μL of sterile water. Samples were sub-jected to positive electrospray ionization (ESI) and detected via multiple reaction monitoring (MRM) using a LC-MS/MS system (Agilent Technologies 6460 QQQ/ MassHunter). Calibration standards were prepared with an inter- and intra-day precision and accuracy of ≤5% with an $r^2$ value of 0.9988 ± 0.0009. Quantification was performed using MRM of the transitions of $m/z$ 172.2 → 128.2 and 176.2 → 128.2 for MNZ and MNZ-d4 respectively. Each transition was monitored with a 100-ms dwell time. Stock solutions of MNZ and MNZ-d4 were prepared at 1 mg/ mL in acetonitrile-water and stored at −20 °C. Mobile phase A is 0.1% acetic acid in $H_2O$ and mobile phase B is 0.1% acetic acid in ACN, and chromatographic separation was achieved using a gradient elution with a Chromolith Performance RP-C18 column maintained at 25 °C from 0 to 4.6 min, B% 0–100, with 0.5 μL/min flow. During pre-study validation, calibration curves were defined in multiple runs on the basis of triplicate assays of spiked media samples as well as QC samples. This method was validated for its sensitivity, selectivity, accuracy, precision, matrix effects, recovery, and stability. Replicates of reference samples were included every six samples and evenly distributed throughout the MS analysis to monitor con-sistency and performance and to utilize for downstream normalization.

**Bacterial quantification.** Bacterial quantification determined via turbidimetry was completed by measuring the optical density at each time point, 100 μL of sample inoculum was read at O.D. 600 nm using a SpectraMax Plus 384 UV

spectrophotometer. Time points were recorded within 5 min of sampling and stored at 4 °C.

Bacterial quantification using plate counting was done by doing a 10-fold dilution using sterile water and aliquoting 100 µL spread evenly onto BD agar plates. Cultures were incubated at 37 °C. Plating was done in triplicates and was counted manually. Prior optimization ensured the dilution would result in no more than 300 colonies making quantification as accurate as possible.

For co-culture validation experiments, 100 µL of sample was aliquoted on Rogosar agar and *Gardnerella* selective agar. Experiments were conducted to verify the absence of Lactobacillus growth on *Gardnerella* selective media and absence of *G. vaginalis* growth on Rogosan agar, to confirm that colony formation specific to respective taxa. Cultures were incubated at 37 °C, with a total of 36 biological replicates for the 1000× and 0.001 Gv:Li ratio cultures ($n = 18$ cultures for each ratio). Plating was done in triplicates and was counted manually. Prior optimization ensured the dilution would result in no more than 300 colonies making quantification as accurate as possible.

**Bacteria–MNZ experiments**. For the MNZ experiments, 50 µL MNZ was added at appropriate concentrations to 5 mL of NYCIII media. Samples equilibrated at 37 °C for 1 h prior to the addition of 50 µL of bacterial inoculum ($2 \times 10^6$). One hundred and fifty microliters of aliquot was taken for time-point readings for MNZ and bacterial quantification (as described above). Samples were incubated at 37 °C under constant mixing and only removed for time-point measurements.

For the co-culture experiments, Gv:Li ratios were added at appropriate experimental conditions in a likewise manner. For each varying ratio sample within each experiment, a side-by-side duplicate was performed without MNZ as a negative control. The negative control was assessed only for bacterial quantification to ensure that no growth condition or external stimuli promoted the growth of one over another. Negative control experiments demonstrated bacterial proliferation that modeled growth curves of each individual bacterium cultured alone thus confirming any changes in growth seen in our bacteria–MNZ experiments were the result of the addition of MNZ.

**ODE models**. The model equations were constructed assuming both *Li* and *Gv* internalize MNZ at rates $k_{int-Li}$ and $k_{int-Gv}$, MNZ toxicity to *Li* and *Gv* occurred at rates dependent on the maximum rates $k_{kill-Li}$ and $k_{kill-Gv}$ and the concentration of internalized MNZ where growth inhibition increased as internalized MNZ exceeded a threshold as described by 50% effective concentrations, $EC50_{Li}$ and $EC50_{Gv}$. The growth of *Li* and *Gv* was assumed to be logistic in behavior at rates $k_{grow-Li}$ and $k_{grow-Gv}$ with distinct carrying capacities for each bacterium, $K_{Li}$ and $K_{Gv}$. The parameters for $k_{grow-Li}$, $k_{grow-Gv}$, $K_{Li}$, and $K_{Gv}$ were determined by nonlinear least-squares fitting of the logistic function to growth curves for *Li* and *Gv* grown in separate cultures (Supplementary Fig. 2a, b)[64]. The $k_{kill-Li}$, $k_{kill-Gv}$, $EC50_{Li}$, and $EC50_{Gv}$ were determined by fitting the Hill equation to kill curves for *Li* and *Gv* cultured in isolation (Supplementary Fig. 2c, d). Internalization rates, $k_{int-Li}$ and $k_{int-Gv}$ and metabolism rates, $k_{acet}$ and $k_{met}$ were determined from fitting the ODE model to time course mass spectrometry data for external MNZ, internal MNZ and acetamide and cell densities (optical density) using a multistart local optimization strategy (*Multistart*) with the local solver *lsqcurvefit*.

**Model simulations and validation**. Unless otherwise noted, all simulations were completed at MNZ concentration of 500 µg/ml over the course of 48 h. Growth outputs were normalized to the maximal growth density ($K_{Li}$ and $K_{Gv}$) for comparison across simulations and to experimental data. External MNZ, internal MNZ, and acetamide concentrations were relative to the total volume of cellular pellets. Sensitivity analyses were completed by perturbing a single model parameter while keeping the rest of the parameters constant over 1000×–0.001× the original value. Surfaces were generated over three orders of magnitude for MNZ concentration (10–1500 µg/ml) and eight orders of magnitude for ratio of Gv:Li ($1.6 \times 10^{-4}$–$1.6 \times 10^4$) at 1225 combinations of MNZ concentration and Gv:Li ratio. Model validation was completed by comparing the experimental co-culture data to model predictions using unpaired *t*-tests.

**Generalized models and global sensitivity analysis**. To incorporate intraspecies and interspecies variation we developed three additional model structures and ran simulations with randomized parameter sets to determine if the influence of initial Gv:Li ratio, or the more generalized BV:LB ratio, on endpoint *Lactobacillus* spp. composition is consistently observed across model structures.

For capturing intraspecies variation, we used Latin Hypercube Sampling of parameter ranges for each parameter to create 100 parameter sets. We derived these parameter ranges from the literature and a summary of these ranges can be found in Supplementary Tables 2 and 3. These same parameter ranges and sampling methods were used for the global sensitivity and uncertainty analysis, which analyzed the partial rank correlation coefficient with 2000 randomly generated parameter sets with endpoint (48 h, 500 µg/ml MNZ) *Lactobacillus* spp. relative abundance[65]. For capturing interspecies variation, and microbe–microbe interactions like cross-feeding, we developed a four species model that includes two representative BV-associated bacteria, and two *Lactobacillus* species, *L. iners* and a second species representing *L. crispatus*, *L. jensenii*, or *L. gasseri*.

*Internalization/uptake rates ($k_{int}$)*. To our knowledge, this is the first publication that demonstrates that *G. vaginalis* and *L. iners* uptake or sequester MNZ. Previous literature describing uptake of MNZ in other bacterial species, including both obligate and facultative anaerobes has been published by Ralph and Clarke[31], Tally et al.[27], and Narikawa[66]. These publications demonstrate that even bacteria that are resistant to MNZ can still uptake MNZ, and at similar rates. Despite the fact that facultative anaerobes are believed to be largely insensitive to MNZ, Narikawa specifically demonstrates that nitroreductase activity is associated with the ability to uptake MNZ, and that pyruvate:ferrodoxin activity is associated with sensitivity to MNZ as an explanation for why the facultative anaerobes *Escherichia coli*, *K. pneumoniae*, *M. morganii*, and *S. faecalis* exhibited high MICs, but reduced supernatant MNZ. We calculated the rates of MNZ uptake for five species, one obligate anaerobe, *B. fragilis*, and four facultative anaerobes (*E. coli*, *S. aureus*, *P. morganii*, and *S. faecalis*) by digitizing the kinetic data for cell counts and extracellular MNZ concentrations in Ralph and Clarke[31] and fitting second-order reaction kinetics by ordinary least-squares regression. The rates ranged from $2 \times 10^{-17}$ to 0.15 cell density$^{-1}$ h$^{-1}$. To determine the likelihood that these parameters could be a basis for *Lactobacillus* spp. uptake of MNZ, we assessed the similarity between *E. coli*'s oxygen-independent NADPH-nitroreductase, nsfA, with nitroreductase protein sequences of *G. vaginalis* (34.7%), *L. crispatus* (31.0%), *L. iners* (29.4%), *L. jensenii* (19.4%), and *L. gasseri* (18.52%). In addition, Guillen et al.[67] reported that *L. plantarum* had selective nitroreductase activity, that shared 32–43% sequence similarity with several *Lactobacillus* species, and in comparison had similarity with *G. vaginalis* (24.0%), *L. crispatus* (38.5%), *L. iners* (25.5%), *L. jensenii* (52.8%), and *L. gasseri* (30.0%). Sequence similarity was assessed by NCBI's protein BLAST[68]. As obligate anaerobes were observed to uptake MNZ at higher rates, we assumed that the other BV-associated bacteria, which could be an obligate anaerobe could potentially have higher capacity to internalize MNZ.

*Growth rates ($k_{grow}$) and carrying capacities ($K$)*. To account for potential variability in growth rates, we surveyed previously published to determine ranges in growth. For *Lactobacillus* species, we calculated growth rates by digitizing growth curves from Chetwin et al.[69] and analyzed growth rates reported in Juárez–Tomás[70], Anukam and Reid[45]. *G. vaginalis* and other bacterial strains growth curves were less abundant in the literature, but we did calculate growth rates from Atassi et al.[49] and Anukam and Ried[45]. Generally, *G. vaginalis* and other BV-associated bacteria seemed to have slower growth rates than *Lactobacillus* species, and in the same culture conditions this was observed in Anukam and Ried[45]. For carrying capacity we assumed that there were similar carrying capacities for all species, except the BV-associated bacteria based on data from Castro et al.[71], that reported *A. vaginae* at lower levels that *G. vaginalis* at steady state[71].

*Sensitivity to MNZ (EC50 and $k_{kill}$)*. MNZ is highly variable, and typically obligate anaerobes are considered the most sensitive to MNZ. The strain of *G. vaginalis* used in the basis of this model is relatively resistant to MNZ, with growth barely inhibited at 256 µg/ml (9% inhibition compared to 0 µg/ml control, Supplementary Fig. 2d). For *A. vaginae*, the MIC can range 2–256 µg/ml and *G. vaginalis* can range from 0.75 to >500 µg/ml[43,44]. Generally, it is assumed that *Lactobacillus* spp. are insensitive to MNZ; however, this also appears to be highly strain and species-dependent with some *Lactobacillus* isolates in similar ranges of sensitivity as *G. vaginalis*[42,45]. The rate at which MNZ inhibits growth is more difficult to find, as the experiments to determine this rate are more laborious than the standard kill curve to calculate EC50 so we assumed all kill rates to be equal across all species.

*Metabolism of MNZ*. To our knowledge, this is the first manuscript to describe the metabolism of MNZ by vaginal microbiota. We solely based the parameter value on the rate observed for the *G. vaginalis* strain from the model. In addition, we assumed that only BV-associated bacteria metabolism MNZ based on the observation that only BV-associated bacteria metabolize HIV microbicide drugs[29].

*Interspecies interaction terms*. Gause[72] noted the calculation for interaction terms for a generalized Lotka–Volterra model describing competitive exclusion (Eqs. (1) and (2)). In our model, we generalized the interaction terms further to be able to capture many different interactions, specifically amensal behavior where *Lactobacillus* spp. can inhibit BV-associated bacterial growth with no effect of BV-associated bacteria on *Lactobacillus* species growth ($-/0$) as well as mutualistic (both species benefit from the other $+/+$) and commensal behaviors (one species benefits $0/+$) between BV-associated bacteria or within the *Lactobacillus* population. The amensal behavior between *Lactobacillus* species has been documented experimentally in co-culture[73] and we calculated the interaction term for many different species and strains of *Lactobacillus* on *G. vaginalis* and *Prevotella bivia* from Atassi et al.[50]. It is largely believed that D-lactic acid produced by many *Lactobacillus* species inhibits the growth of BV-associated bacteria; however, *L. iners* does not produce this isomer of lactic acid and is the reasoning behind not including an interaction term between *L. iners* and the BV-associated bacteria[48,74]. It is believed that commensal behavior exists between *G. vaginalis* and *P. bivia* in the form of cross-feeding, so we allowed the model to simulate this behavior[51]. In addition, *G. vaginalis* is associated with promoting the growth of other BV-associated bacteria like *A. vaginae*[71]. Calculations were completed assuming the reported mono and co-cultures were at steady state to derive Eqs. (3) and (4).

Equations (3) and (4) relate to the parameters in Supplementary Table 3 by Eqs. (5) and (6), which generalizes the reported interaction strength from the literature to be able to be adjusted for varying carrying capacities simulated in the model that do not equal the carrying capacities from the literature.

$$\frac{dN}{dt} = r_N N \left[ 1 - \frac{N + s_{P \to N} P}{K_N} \right] \qquad (1)$$

$$\frac{dP}{dt} = r_P P \left[ 1 - \frac{P + s_{N \to P} N}{K_P} \right] \qquad (2)$$

$$s_{P \to N} = \left[ \frac{K_N - N}{P} \right] \qquad (3)$$

$$s_{N \to P} = \left[ \frac{K_P - P}{N} \right] \qquad (4)$$

$$s_{P \to N} = \left[ \frac{K_N - f_{P \to N} K_N}{f_{N \to P} K_P} \right] \qquad (5)$$

$$s_{N \to P} = \left[ \frac{K_P - f_{N \to P} K_P}{f_{P \to N} K_N} \right] \qquad (6)$$

*Software.* Parameterization, ODE modeling, and sensitivity analyses were completed using Matlab® 2018b (Matlab, Natick, MA). Statistical analyses were performed using PRISM 8, exact *p*-values <1E–6 were calculated in Matlab.

### Clinical data and study population

*The UMB-HMP cohort.* The study results and associated clinical data were previously published[33] and all data provided were de-identified to this study. The UMB-HMP study was not an interventional study, but an observational prospective study, where treatment information was recorded during a clinical exam at week 5 and week 10 for 135 nonpregnant women of reproductive age. Within this study, MNZ treatment was provided as standard of care, as recommended by the CDC (metronidazole 500 mg orally twice a day for 7 days)[75]. The original study protocol was approved by the Institutional Review Board of the University of Alabama at Birmingham and the University of Maryland School of Medicine. Written informed consent was appropriately obtained from all participants, who also provided consent for storage and use in future research studies related to women's health.

Women self-collected cervicovaginal swabs for 10 weeks. Vaginal microbiota data were generated by sequencing the V3–V4 regions of the 16S rRNA gene and is available in dbGAP BioProject PRJNA208535. In this study, the vaginal microbiota composition data from 11 women who experienced BV and were treated with MNZ during the UMB-HMP study were analyzed. Any participants who failed to complete the MNZ regimen, who did not have BV according to Nugent scoring at the time of MNZ treatment, or who did not have follow-up data available were excluded from the analysis. The initial relative abundances were averaged across the week before starting MNZ treatment. Patients were classified to have recurrent BV if they exhibited a second episode of BV based on Nugent scoring (7–10) during remaining of the 10-week observation period.

*The CONRAD BV cohort.* The study results and associated clinical data were previously published[34] and all data provided were de-identified to this study. The original clinical study protocol was approved by the Chesapeake Institutional Review Board (IRB) (Pro #00006122) with a waiver of oversight from the Eastern Virginia Medical School (EVMS) and registered in ClinicalTrials.gov (#NCT01347632). A total of 69 women were screened from symptomatic discharge and 35 women were enrolled in the study. Vaginal microbiota data were generated by sequencing the V4 region of the 16S rRNA gene, providing taxonomic resolution at the general level.

Thirty-three women completed all three visits. BV was evaluated by vaginal microbiota compositional data (molecular-BV)[76]. After biological samples were obtained at visit 1 (V1), women with BV were prescribed twice daily, 500-mg MNZ for 7 days. Participants returned for visit 2 (V2) 7–10 days after completing the course of MNZ therapy and visit 3 (V3) 28–32 days after completing treatment. At all three visits, samples were obtained to evaluate vaginal semen (ABAcard, West Hills, CA), vaginal pH, gram stain for Nugent score and semiquantitative vaginal flora culture. CVLs were collected, followed by vaginal swabs and three full-thickness biopsies.

### Analysis of clinical outcomes

In the Human Microbiome Project cohort, patients were defined as cured or recurrent based on whether after initial MNZ treatment the patient suffered an additional episode of BV (Nugent 7–10) during the 10-week course of data collection. For analysis, initial flora relative abundances were averaged across the 7 days prior to reported treatment start date. To analyze the relative ratio between BV-associated bacteria and *Lactobacillus* spp., we combined the relative abundances for the top 20 BV-associated bacteria and all *Lactobacillus* spp. The genera BV-associated bacteria included were *Gardnerella, Atopobium, Megasphaera,* BVAB1-3, *Streptococcus, Prevotella, Leptotrichia, Anaerococcus,*

*Peptoniphilus, Eggerthella, Veillonella, Sneathia, Mobiluncus, Corynebacterium, Ureaplasma, Eubacterium, Porphyromonas, Dialister, Peptostreptococcus, Bacteroides, Fusobacterium, Actinomyces,* and *Bifidobacterium.* Before statistical analysis, the BV:LB ratio was log-transformed, and the relative abundances of *L. iners, G. vaginalis* were center-log ratio (CLR) transformed, with pseudocounts added to taxonomic units with relative abundances equal to zero. Statistical analysis of the BV:LB ratio and Gv:Li ratio was completed using two-sided unpaired Student's *t*-tests and analysis of the CLR-transformed single species abundances was completed using two-sided unpaired Student's *t*-tests and were corrected using the FDR method of Benjamini and Hochberg (PRISM 8).

For the CONRAD BV cohort, treatment outcome was defined based on *Lactobacillus* dominance evaluated at enrollment, 7 days after treatment and 28–32 days after treatment. Patients that exhibited *Lactobacillus* dominance at both 1 week and 1 month after treatment were considered cured, and patients that exhibited *Lactobacillus* dominance only at week 1 and not at 1 month were considered recurrent. The statistical analysis followed the same methodology as the HMP cohort.

**Reporting summary**. Further information on research design is available in the Nature Research Reporting Summary linked to this article.

## Data availability

The source data are provided with this paper for Figs. 2–5, Supplementary Figs. 1, 2, 4 and 8, which includes the in vitro validation, model parameterization, and clinical validation. For the clinical studies, the UMB-HMP cohort study sequence data and metadata were deposited in the Sequence Read Archive (SRA; http://www.ncbi.nlm.nih.gov/Traces/sra/) under BioProject PRJNA208535 (The daily dynamics of the vaginal microbiota before and after bacterial vaginosis) ([SRP026107] and [SRA091234])[33]. An abbreviated data set necessary for the reproduction of Fig. 5a, b and Supplementary Fig. 8a, b are in Supplementary Table 4 and the Source data file. The sequence data and metadata for the CONRAD BV study are not in a formal repository, but are fully available upon request; however, we have included an abbreviated version of this data set that includes all the data necessary for reproduction of Fig. 5c, d and Supplementary Fig. 8c, d are in Supplementary Table 5 and can be found in the Source data file. Source data are provided with this paper.

## Code availability

The code used to generate the model simulations is published (https://doi.org/10.5281/zenodo.4121904)[77].

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

## Acknowledgements

This work was in part funded by startup funds from the University of Miami to N.R.K., funds from NIH/NIDDK grant RO1DK112254 to N.R.K., and NIH/NIAID grant R01AI138718 subcontract to N.R.K., and startup funds from the University of Michigan to K.B.A. The UMB-HMP study, J.R. and M.F. were supported by the National Institute for Allergy and Infectious Diseases of the National Institutes of Health under award numbers UH2AI083264 and R01NR015495. The CONRAD BV Study was funded by an intra-agency agreement between the Centers for Disease Control and Prevention (CDC), United States Aid and International Development (USAID) and CONRAD (GPO-A-00-08-00005-00). The views expressed by the authors do not necessarily reflect those of the funding agency or CONRAD.

## Author contributions

C.Y.L., R.K.C., M.M.L., N.R.K., and K.B.A. conceived and designed the study. C.Y.L. completed the computational analysis and analyzed the clinical data. R.K.C., N.R.K., and B.H. designed and conducted monoculture and co-culture kinetic experiments. A.G., M.F., and J.R. curated data from the UMB-HMP cohort. J.R. led the UMB-HMP study and data collection. A.T. and G.D. provided CONRAD BV protocol development, patient care, and data analysis. C.Y.L., R.K.C., K.B.A., and N.R.K. wrote the manuscript and all authors read and revised the manuscript.

## Competing interests

J.R. is co-founder of LUCA Biologics, a biotechnology company focusing on translating microbiome research into live biotherapeutic drugs for women's health. All other authors declare no competing interests.
