## [Peer Review File · Nature Communications]

REVIEWERS' COMMENTS

Reviewer #1 (Remarks to the Author):

I reviewed earlier version of this paper for another journal. Though I remain enthusiastic about the topic, my main concern regarding the added-value of the modelling (given that the accumulation had been described before) remains open. The new modelling results go only a little towards this, and experimental validation is lacking.

Reviewer #2 (Remarks to the Author):

This manuscript presents a novel systems-based model for the identification/prediction of women who will experience recurrent BV after metronidazole treatment. The authors have responded appropriately to the concerns of the review. The results of these types of studies may eventually have clinical applications.

Reviewer #3 (Remarks to the Author):

I believe the authors addressed the concerns raised in previous reviews thoughtfully and in detail, and recommend acceptance.

REVIEWERS' COMMENTS

Reviewer #1 (Remarks to the Author):

I reviewed earlier version of this paper for another journal. Though I remain enthusiastic about the topic, my main concern regarding the added-value of the modelling (given that the accumulation had been described before) remains open. The new modelling results go only a little towards this, and experimental validation is lacking.

We agree that the potential for non-antibiotic-target bacterial populations to act as a sink for MNZ and alter efficacy is similar to a concept that has been previously explored in bacterial ecology, termed the inoculum effect (IE), which describes an increase in antibiotic MICs due to increased bacterial load and decreased per cell antibiotic concentration.⁵⁴ While the IE and the ability of bacterial species to influence MNZ bioavailability has indeed been previously reported, to our knowledge its association with vaginal *Lactobacillus iners* and role in BV treatment failure has not yet been considered. The ODE model used here was essential for determining the critical importance of MNZ sequestration by *Lactobacillus* spp. across multiple interactions that have the potential to influence efficacy and recurrence, including metabolism, proliferation, and susceptibility to MNZ of both target and non-target species. The model was also necessary for translating the importance of this parameter to microbial communities with varying compositions and with different MNZ dosing regimens. The model result that more *Lactobacillus* spp. present at treatment initiation could negatively MNZ efficacy is unexpected given that the goal of therapy is to shift the community back toward *Lactobacillus* dominance.

Though the proposed MNZ sequestration mechanisms were not experimentally validated in this study, the model predictions for associated relationships between pre-treatment microbial composition and BV recurrence were recapitulated in both co-cultures and in cervicovaginal samples. We do acknowledge that direct experimental validation of the MNZ sequestration mechanism is a priority for future work.

We thank the reviewer for these comments and have tried to better clarify these points in the Introduction (p. 5) and Discussion (p. 18).

Reviewer #2 (Remarks to the Author):

This manuscript presents a novel systems-based model for the identification/prediction of women who will experience recurrent BV after metronidazole treatment. The authors have responded appropriately to the concerns of the review. The results of these types of studies may eventually have clinical applications.

We thank the reviewer for careful evaluation of the previous revisions.

Reviewer #3 (Remarks to the Author):

I believe the authors addressed the concerns raised in previous reviews thoughtfully and in detail, and recommend acceptance.

We thank the reviewer for careful evaluation of the previous revisions.